# Crop Rotation and Minimal Tillage Selectively Affect Maize Growth Promotion under Late Wilt Disease Stress

**DOI:** 10.3390/jof8060586

**Published:** 2022-05-30

**Authors:** Ofir Degani, Asaf Gordani, Paz Becher, Assaf Chen, Onn Rabinovitz

**Affiliations:** 1Plant Sciences Department, Migal–Galilee Research Institute, Tarshish 2, Kiryat Shmona 11016, Israel; asigordani1@gmail.com (A.G.); pazbec@gmail.com (P.B.); onnrab@gmail.com (O.R.); 2Faculty of Sciences, Tel-Hai College, Upper Galilee, Tel-Hai 12210, Israel; assafc@migal.org.il; 3Soil, Water, and Environment Department, Migal–Galilee Research Institute, Tarshish 2, Kiryat Shmona 11016, Israel

**Keywords:** *Cephalosporium maydis*, crop cycle, crop protection, disease control, fungus, *Harpophora maydis*, late wilt, *Magnaporthiopsis maydis*, maize, real-time PCR

## Abstract

In recent years, worldwide scientific efforts towards controlling maize late wilt disease (LWD) have focused on eco-friendly approaches that minimize the environmental impact and health risks. This disease is considered to be the most severe threat to maize fields in Israel and Egypt, and a major growth restraint in India, Spain, and Portugal. Today’s most commonly used method for LWD control involving resistant maize genotypes is under constant risk from aggressive pathogen lines. Thus, this study’s objectives were to evaluate the effect of crop rotation and avoiding tillage on restraining the disease. Such an agrotechnical approach will support the continuity of soil mycorrhiza networks, which antagonize the disease’s causal agent, *Magnaporthiopsis maydis*. The method gained positive results in previous studies, but many knowledge gaps still need to be addressed. To this end, a dual-season study was conducted using the LWD hyper-susceptible maize hybrid, Megaton cv. The trials were performed in a greenhouse and in the field over full dual-growth seasons (wheat or clover as the winter crop followed by maize as the summer crop). In the greenhouse under LWD stress, the results clearly demonstrate the beneficial effect of maize crop rotation with clover and wheat on plant weight (1.4-fold), height (1.1–1.2-fold) and cob yield (1.8–2.4-fold), especially in the no-till soil. The clover-maize growth sequence excels in reducing disease impact (1.7-fold) and pathogen spread in the host tissues (3-fold). Even though the wheat-maize crop cycle was less effective, it still had better results than the commercial mycorrhizal preparation treatment and the uncultivated non-infected soil. The results were slightly different in the field. The clover-maize rotation also achieved the best growth promotion and disease restraint results (2.6-fold increase in healthy plants), but the maize rotation with wheat showed only minor efficiency. Interestingly, pre-cultivating the soil with clover had better results in no-till soil in both experiments. In contrast, the same procedure with wheat had a better impact when tillage was applied. It may be concluded that crop rotation and soil cultivation can be essential in reducing LWD, but other factors may affect this approach’s benefits in commercial field growth.

## 1. Introduction

Maize (*Zea mays* L., corn) late wilt disease (LWD) has been investigated extensively in Israel in the past decade in order to improve our toolkit in order to study its causal agent, *Magnaporthiopsis maydis*, biology and pathogenesis [1]. This fungal pathogen was known in the past under the synonyms *Cephalosporium maydis* [2] and *Harpophora maydis* [3,4]. Such efforts were part of a larger goal—improving existing methods and identifying new ones to eliminate or repress this destructive threat [5]. Indeed, Israel and Egypt are considered the world’s most LWD-affected areas, perhaps because the disease first evolved in this area [2,6,7], or due to favorable environmental conditions that enable the pathogen to thrive [8]. Other countries affected by LWD are Spain, Portugal [9], India [10] and Hungary [11]. This world distribution could expand due to seed transmission and global warming.

When a susceptible maize cultivar is seeded on infected soil, the pathogen enters the plant’s roots and spreads gradually upwards in the vascular system (the xylem) [12,13]. This establishment phase occurs almost without visible signs. Yet careful examination of sprouts in their first three weeks of living can reveal the damage to the development and health of the roots’ system [7,14]. Later, at the male flowering phase (ca. 50 days from sowing), the plants’ lower stem and leaves begin to show dehydration stress symptoms [13,15]. As the growing season progresses, the fungus spreads upwards to the upper parts and eventually to the cobs [16]. Indeed, the most globally concerning LWD dispersal mode is seed-borne [17], but dispersal through infested soils and agriculture equipment become crucial when local quarantine is needed [18].

Moreover, the pathogen can survive for long periods as sclerotia or spores in the soil [19,20] and as a hidden endophyte in secondary host plants [21,22,23]. Although LWD appears in many cases as patches scattered in the field [24], LWD may result in total field infection and yield loss in heavily infected areas planted with highly susceptible maize cultivars [25,26]. A similar asymptomatic infection mode occurs in resistant cultivars with some delay and can result in infected seeds that enhance the pathogen’s spread to a new area [16,25].

Therefore, developing control methods against the LWD fungus is an urgent need. Indeed, various control methods have been suggested to cope with LWD, some of which showed promising results in reducing the disease outcome in commercial fields. These methods include watering the field [27], balanced soil fertility [28,29], soil solarization [30], allelochemical [14], biological [31,32,33,34,35,36,37,38], and chemical options [39,40]. Lately, new antifungal compounds (such as the resistance inducer beta-sitosterol, chitosan nanoparticles, and silica nanoparticles) have been suggested against the LWD pathogen [38,41,42]. An economical, efficient, and practical Azoxystrobin-based control protocol [25,26,43,44] was proposed for the commercial protection of LWD-susceptible maize genotypes under heavy pathogen load stress. Yet, since fungicide treatment limitation is exerting increasing pressure in many countries due to environmental and potential health risks, searching for green alternatives to cope with LWD is a continuous worldwide effort. Many studies have aimed at LWD biological control [31,32,33,45,46] to address this challenge, including direct intervention using antagonistic bacteria and fungi or their secreted metabolites [34,47]. An alternative approach is to operate and strengthen beneficial microorganism communities in the soil (for example, by compost addition [48]).

While many of these studies showed promising results, today’s most commonly used method for LWD control relies on developing genetically resistant maize cultivars [12,49,50,51,52]. This choice of disease control is not without drawbacks. The discovery of highly aggressive *M. maydis* isolates [53,54,55] poses a new challenge to researchers. These virulent fungal varieties may threaten resistant maize cultivars, specifically when growing resistance genotypes in the exact location for extended periods [24,26]. This alarming situation is encouraging researchers to continue to seek new methods to control LWD.

It was recently demonstrated in two separate studies in Portugal [35] and Israel [56] that the tillage system, cover crop, and crop rotation could be promising agrotechnical methods for suppressing LWD. Selected cover crops, crop cycle, or no-till cropping could assist in building mycorrhizal networks that function throughout a sequence of several main crops [57]. Such cultivation practices can restrain *M. maydis*’s establishment and plant penetration phases (summarized by [36]). Maintaining the integrity and continuity of mycorrhizal communities in the soil has already proved essential in the plant’s ability to cope with soil biotic and abiotic stresses by activating the plant’s local and systemic defense mechanisms [58,59]. Indeed, Arbuscular mycorrhizal fungi (AMF) have a well-studied rule in improving plant resistance to diseases [60]. They also play a pivotal role in plant nutrition [61].

Until recently, the benefits of strengthening the soil microbiome were poorly tested against the late wilt pathogen. The first study examining this approach (Patanita et al., 2020, [35]) showed that the presence of *M. maydis* was significantly reduced, and grain production improved considerably when cover crop and minimum tillage were applied together. It was also found that arbuscular root colonization was higher with cover crop and minimum tillage. In a previous study [56], we tested the effect of crop cycle and no-till on LWD in a greenhouse using the susceptible maize hybrid, Prelude. When maize was seeded after wheat cropping, a significant enhancement in the shoot’s fresh weight (47–54%) and cob (36–46%) was accomplished compared to the other conditions (clover soil, commercial mycorrhiza preparation and bare soil control). This achievement was not affected drastically by tillage. It was followed by a sharp decrease in disease symptoms (73%) and the pathogen’s presence (64–82%) in the plants’ tissues. It was concluded that, since wheat and maize are more closely related (they are both *Poaceae*) than clover and maize, they might share similar mycorrhizal networks adapted to perform better in these crops.

The above findings encourage further exploration of crop rotation (clover/wheat) with maize and the effect of tilling on the impact of LWD on maize development and yield. It is vital to see if the maize hybrid selection (exceptionally high susceptible lines) and field conditions may alter the outcome of those cultivation practices. This method was examined thoroughly over double cultivation and two seasons of study in the current work. Crop rotations of clover/maize or wheat/maize and tillage regimes (no-tillage or conventional tillage) were examined in greenhouse pots and in the field under LWD stress using the hyper-susceptible maize cultivar, Megaton cv. These treatments were compared to a commercial mycorrhizal preparation (Resid MG [62], Mico, BioBee, Sde Eliyahu, Israel), and infected and non-infected bare soil controls. The efficiency of these agrotechnical practices on *M. maydis* pathogenesis was evaluated by following the plants’ growth and yield parameters, estimating disease severity, and quantifying the pathogen’s DNA inside the plants’ tissues using a real-time (qPCR)-based technique.

## 2. Materials and Methods

### 2.1. Rationale and Timetable

This study inspected the effect of crop cycling and land processing on LWD outbursts over a whole double-cultivation season in greenhouse pots and in the field. A binary rotation (with 20 days entry phase) of clover–maize or wheat–maize was studied during the fall and winter until the summer of 2021. The use of pots provides more uniform environmental conditions and better isolation of each treatment’s influence. It is important for controlling soil composition, inoculation and water regime. It also enables us to add a negative control—non-infected plants. Therefore, it is an essential step designed to complete the image, along with the field experiment conducted in parallel. Together, the two experiments provide a better understanding of the variables associated with the former crop, soil cultivation and the environment. The two experiments’ timetables are described in Table 1. The field experiment began shortly after the greenhouse experiment for practical reasons (mainly the availability of the field). All measurements included at least 10 independent biological replications.

### 2.2. Crop Rotation and Tillage Impact in Pots in a Greenhouse

#### 2.2.1. Fungal Source and Growth Conditions

The *Hm-2* isolate of *M. maydis* (CBS 133165, CBS-KNAW Fungal Biodiversity Center, Utrecht, The Netherlands) was isolated from infected corn plants collected from Sde Nehemia (northern Israel, Upper Galilee, Hula Valley) in 2001 and identified as previously described [16,63]. Colonies were grown on rich potato dextrose agar (PDA) (Difco, Detroit, MI, USA) solid medium in an incubator at 28 ± 1 °C in the dark for 4–7 days.

#### 2.2.2. Infected Sterilized Wheat Grains Preparation

Infected sterilized wheat grains were used to spread *M. maydis* in the soil and assure a high inoculation load, as previously described [64]. Wheat grains were maintained overnight in tap water, then dried in a fume hood on paper towels for 5 h. The dried seeds were then sterilized by autoclave for a half-hour at 120 °C. In disinfected plastic boxes (0.5 L volume), 150 g sterilized wheat grains were incubated with 20 *M. maydis* colony agar discs. The colony agar disks (each 6 mm in diameter) were taken from the borders of a young (4–6-day-old) fungal colony grown in the dark at 28 ± 1 °C. The boxes’ lids were tightened using Saran wrap, and each box was covered with aluminum foil (for dark conditions) and incubated at 28 ± 1 °C in the dark for 10–14 days. The seeds were mixed once every two days by shaking to ensure uniform inoculation.

#### 2.2.3. The Plants’ Infection Technique

The maize growth and plant infection methodologies were similar to previous studies [21,25] and included two steps:[1].Inoculation I: the use of a field soil having a history of LWD infection (Gadash Experimental Farm located near the Gome junction, Hula Valley, Upper Galilee, northern Israel, 33°10′48.6″ N 35°35′11.6″ E) [32].[2].Inoculation II: to ensure moist soil conditions, all pots were irrigated one day prior to this step. Sterilized *M. maydis*-infected wheat seeds (150 g, prepared as described in Section 2.2) were mixed with the top 20 cm of the pot’s soil ca. three week before the winter season experiment. The soil was kept moist until the clover and wheat were sown.

#### 2.2.4. The Greenhouse Trial Protocol

The greenhouse experiment was conducted at the R&D North Israel Experimental Farm located in Upper Galilee, Hula Valley, northern Israel (33°09′08.2″ N 35°37′21.6″ E). This study included two treatments, commercial control, positive control, and negative control. The treatments included crop cycle with clover or wheat (planted before maize) in dual cultivation with or without tillage, as described in Figure 1. Controls without winter cropping included the addition of commercial mycorrhiza preparation, infected control (naturally infested soil with the addition of complementary *M. maydis* infection), and non-infected soil.

The commercial mycorrhizal treatment was Resid MG [62] (or Mico) preparation (manufactured by Symborg S.L., Murcia, Spain, supplied by BioBee Biological Systems, Sde Eliyahu, Israel). This product contains clay particles, 1.6 × 10^4^ spores/kg of the Arbuscular mycorrhizal, *Glomus iranicum* var. *tenuihypharum*. According to the manufacturer’s directions, 5 g of the product powder was added to each seedling one-week post-sowing. For the negative control, local peat soil was collected from the experimental farm’s field. This control soil is similar to the infected soil collected ca. 4 km nearby. It has no known history of LWD infestation (if such infestation exists, it is estimated to be minor). Each of these treatments was carried out twice, once with soil cultivation (which simulates conventional tillage) and once with minimal soil intervention. The soil cultivation was conducted using vigorous mixing and disintegration of the soil by pickaxe.

Each treatment included 10–12 independent repetitions—110 pots in total were scattered in a completely randomized design. The pots were placed on concrete blocks to prevent contact of the roots with the ground. Infected field peat soil (see Section 2.3) was mixed with coarse perlite (No. 4, for aerating the ground) at a ratio of 2:1 in 10-L pots. Each pot was sown with five seeds. All seeds were pre-treated with thiram, captan, carboxin, metalaxyl-M (Rogers/Syngenta Seeds, Boise, ID, USA, supplied by CTS, Tel Aviv, Israel), a standard general pesticide treatment. The seeds were tested for vitality before sowing. Watering was carried out from the seeding stage using 2 L per pot per day. According to the Israel Ministry of Agriculture Consultation Service (SAHAM) growth protocol, insecticide and fertilizer treatments were applied.

The experiment’s detailed timetable is described in Table 1. In the winter season, the clover cultivar was Tabor, an annual single-harvest cultivar popular among Israel’s growers since it is well-adapted to most soils. The wheat cultivar was Zahir, common in Israel’s commercial fields due to its being well-matched for semi-desert conditions and rich in crops even under low rainfall conditions. Hazera Seeds Ltd. (Berurim MP Shikmim, Israel) supplied the wheat and clover cultivars.

The maize Megaton cv. selected for the experiment was produced by Limagrain, Saint-Beauzire, Puy-de-Dôme, France (marketed by Hazera Seeds Ltd.). This variety was previously proven to have very high susceptibility to corn LWD [64]. During the clover/wheat winter growth period, the average greenhouse temperature was 17.7 °C, with a minimum of 5.5 °C and a maximum of 33.7 °C. During the maize summer growth period, the average greenhouse temperature was 24.8 °C, with a minimum of 5.4 °C and a maximum of 41.4 °C. In the maize season, the average greenhouse humidity was 91.3%, with a minimum of 25.9% and a maximum of 100%. Three days from the maize sowing day, an aboveground surface peek assessment was carried out, and 42 days later, the young plants were thinned and one plant was left in each pot. On the thinning day and at the end of the experiment (day 83), the following indices were examined: plant height; shoot fresh weight; phenological stage (leaves’ number); and the severity of dehydration signs. Also, the plants’ survival rate and cobs’ wet weight were determined 45 and 83 days after sowing (DAS), respectively. Wilt valuation was based on the number of dry leaves out of the total leaves and calculation of their percentages, as detailed below:On day 45, we sampled ca. 130–210 leaves from each treatment/control (a total of 1855 leaves were collected). The number of dry leaves was stated in each repeat, and the percentage of those leaves was calculated.On day 83, we sampled 100–120 leaves from each treatment/control (a total of 1121 leaves were collected). The number of dry leaves was stated in each repeat, and the percentage of those leaves was calculated.

In addition, to determine the infection level (pathogen DNA in the plant tissues), DNA was extracted from the roots or first internode tissues of 10 plants per treatment and used in a qPCR test.

### 2.3. Crop Rotation and Tillage Impact in the Field

The field experiment was conducted as similarly as possible to the greenhouse experiment. This includes the same clover, wheat and maize cultivars, the same commercial mycorrhizal preparation (Resid MG), the same soil, and nearly the same growth periods and timetables (see Table 1). The field experiment area was about 0.25 ha in size and it was conducted in five random blocks per treatment (and control) format. Each block included six garden beds arranged in a split-plot design (Figure 2). The garden beds were 6 m wide, 9 m long, and had two rows with 96 cm row spacing. Two land processing regimes were applied in each plot: (1) no-till—the winter crop was harvested, and the clover and wheat stalks remains were left to dry on the soil surface; and (2) conventional tillage. All treatments were sown with hyper LWD-susceptible Megaton cv. maize and the seeds were pre-treated with thiram, captan, carboxin, and metalaxyl-M (as in the greenhouse experiment).

The experiment was conducted at the Gadash Experimental Farm (Upper Galilee, Hula Valley, Northern Israel). The experiment field had a record of moderate LWD infection. In previous studies carried out in 2019 and 2020, we deliberately inoculated the experimental area [32,33]. Thus, no negative control (plots without the pathogen) was included in the experiment. The meteorological data documented during the growing season were ordinary for the winter cropping and nearly optimal (as detailed in [43]) for the LWD burst (Table 2).

#### 2.3.1. Winter Season—Clover and Wheat Cropping

Hazera Seeds Ltd. supplied the wheat (Zahir genotype) and clover (Tabor genotype) cultivars. The field area was not watered during the winter season, because there was enough rain (395.6 mm) to supply the clover and wheat plants’ needs.

#### 2.3.2. Summer Season—Maize Cropping

In the spring of 2021, the plots were sown to a depth of ca. 4 cm with seven plants/meter and germinated the following day using a frontal irrigation system. According to the manufacturer’s directions, 5 g/seed of the Resid MG (or Mico) product powder was added with the sowing. Watering was done using a 20 mm drip irrigation line for each row (Dripnet PC1613 F, Netafim USA, Fresno, CA, USA). It was controlled using a computerized irrigation system at a flow rate of 0.6 l h^−1^. The watering regime was three cubic meters per 0.1 ha/day. The irrigation was adjusted to the plants’ needs (typically every two days). The total water quantity supplied to the field throughout the season was about 400 mm. The plots were treated with fertilization and insecticides at recommended dosages (Consultation Service, SAHAM, Israel Ministry of Agriculture).

Six DAS, the germination percentages were evaluated. At 50 DAS, the plants’ growth indices (shoot wet weight and height, phenological stage) were measured. At the experiment’s end (80 DAS), the plants’ health status and yield (total cob weight and yield quality) of each treatment and the control plots were stated. The yield evaluation included all cobs in the experiment field (15-37 cobs per repeat), so a total of 105-138 cobs were collected for each treatment. The health assessment of the whole plant was based on four categories as previously described [33]: healthy (4); minor symptoms (3); dehydrated (2); and dead (1). We sampled 110–140 plants from each treatment (a total of 987 plants were evaluated). The average plants’ numbers in each disease stage in each treatment were calculated and presented. On both sampling days, samples from the roots (at the sprouting phase, 50 DAS) or the first aboveground internode of the plant stems (at 80 DAS) were taken for DNA purification and qPCR.

### 2.4. Molecular Diagnosis

The qPCR analysis of *M. maydis* DNA in corn plants was made separately to the plants’ roots (in the sprouting phase) or the initial aboveground stem’s internode (in the plants’ maturation phase). The plants’ tissues were washed thoroughly with running tap water, then twice with sterile double-distilled water (DDW). The plant parts were cut into ca. 2 cm a section, and the total weight of each repetition was carefully adjusted to 0.7 g. For DNA isolation and purification, the Murray and Thompson (1980) protocol was used, with slight modifications as described earlier [24]. DNA samples were stored at −20 °C until they were used for the qPCR. This molecular technique, as previously described [25], is based on a typical qPCR method used to detect mRNA (converted to cDNA). It was adjusted for *M. maydis* DNA detection using species-specific primers (Table 3) [4,44]. The A200a primers [16] were used for qPCR. The housekeeping COX gene [65,66] was used as positive control and for normalizing the *M. maydis* pathogen DNA. This gene encodes cytochrome C oxidase (the last mitochondria in the cellular respiratory electron transport chain enzyme). Amplification of the COX gene was done using the primer set COX F/R (Table 3). The relative gene abundance (*M. maydis*/*Cox* DNA ratio) was calculated using the ΔCt model. Similar efficacy was presumed for all samples. All qPCR amplifications were performed in three-four repeats (technical repeats).

### 2.5. Statistical Analysis

A completely randomized design was used in the greenhouse and in the field studies to assess the clover-maize and wheat-maize crop cycles under a tillage/no-till cultivation method to reduce LWD damage. Data statistics were analyzed using the JMP 15th edition program (SAS Institute Inc., Cary, NC, USA). No statistically significant random effect was found in the greenhouse pots’ setting or the field plots’ positioning on the results. Thus, one-way analysis of variance (ANOVA) was applied with a significance threshold of *p* < 0.05. Following the ANOVA analysis, a post hoc of the Student’s *t*-test for all possible individual comparisons (no adjustment for multiple tests) was used. All the data in this study were subjected to the same analysis method. Normally, in field condition experiments, molecular DNA quantities have a high level of variation due to changes in environmental conditions, host susceptibility and the spreading nature of the LWD pathogen [1]. Consequently, fairly high standard error values resulted in most such tests, and statistical significance between the treatments could barely be identified.

## 3. Results

All the results presented below describe the maize growth and health parameters due to the former winter cropping and land processing. As will be detailed below, those agrotechnical approaches yielded a measurable and, in some cases, significant (*p* < 0.05) plant resistance to late wilt disease (LWD). Such statistical differences were easier to detect in the greenhouse, where the conditions were more fixed and optimal. Most growth and health status indices had similar tendencies under greenhouse and field conditions, but some intriguing differences were observed.

### 3.1. Crop Rotation and Tillage Impact in Pots in a Greenhouse

A dual-cropping experiment in a greenhouse throughout a full growing season was performed to examine the effect of winter cultivation (clover or wheat) and soil cultivation (tillage) on the development of late wilt disease on disease hyper-susceptible corn genotype, Megaton cv. The impact of these agrotechnical methods on the germination stage (day 3 from sowing) and the end of the sprouting phase (day 45 from sowing) are shown in Figure 3.

The aboveground maize sprouts’ emergence at 3 DAS (Figure 3A) was only slightly affected by crop rotation or land cultivation. In contrast, an estimate of plants’ growth values made on day 45 of the maize cropping (Figure 3B–D) showed that the uncultivated lands improved plant development, except in the wheat soil and the reduced-infected farm soil control, where the tillage enhanced the growth. However, the wheat soil’s difference between tillage and no-till was statistically insignificant. This is also true for most of the other treatments, indicating a minor tillage system impact at the sprouting phase.

The best result (with statistical significance in the weight and height indices compared to some of the other treatments) was obtained in the clover no-till soil. As expected, the infected control soil left bare throughout the winter (especially if it was processed by tillage) resulted in relatively low (and often the lowest) plant development. Unexpectedly, the commercial mycorrhiza treatment (the Resid MG product application) was also inefficient and yielded similar results to that control. The survival rate at 45 DAS was 58–70% and was approximately equal in most treatments (Figure 3E). Here, too, the infected control and the commercial mycorrhiza treatment resulted in only 48–64% surviving plants.

Surprisingly, the above growth tendencies correlated with the dehydration levels, which were highest in the clover no-till treatment and decreased in the non-cropped infected control and commercial Resid MG treatment (Figure 3F). The increased onset of the disease in plants with accelerated phenological development (e.g., in corn plants grown on uncultivated clover soil) does not necessarily reflect a decrease in plant resistance. This may be due to the phase prior to the appearance of symptoms. To support this explanation, a qPCR analysis revealed that the no-till clover soil led to relatively low pathogen DNA spreading in the plants’ tissues at 45 DAS, which lasted until the season’s end (83 DAS, Figure 4). In this measure (qPCR), the tillage enhanced the pathogen in the clover-maize rotation (on both sampling days) and the non-infected control (at 45 DAS), but not in the other treatments. The wheat soil treatment (regardless of the tillage system) stood out regarding the roots and stems’ first internode high *M. maydis* DNA levels.

At the season’s end, the plant growth (weight and height) and cob yield parameters evidently demonstrate the no-till soil’s advantages (Figure 5). The best development (Figure 5A–C) and yield promotion (Figure 5D) under LWD stress, were reached in that soil when maize was grown in rotation with clover or wheat. The application of commercial mycorrhizal preparation (Resid MG) was beneficial in terms of plant weight and height but did not reflect improved yield production. The phenological development of the plants at harvest was similar in all treatments (9.3–10.3 leaves stage). The wilting percentage estimation (Figure 5E) also resulted in similar non-statistical differences between treatments (30% in the clover soil, 37% in the cultivated wheat soil) with one exception: the no-till bare soil resulted in significantly high (51%) dehydration.

As explained in the Materials and Methods section, for the negative control, local peat soil was collected from the experimental farm’s fields, which had no known history of LWD infestation (if such infestation exists, it is estimated to be minor). This control aimed at achieving as similar as possible soil conditions to the infected treatments’ soil. Yet, despite our expectations, the soil that had not undergone deliberate inaction had some contamination with the pathogen, reflected in the highly sensitive qPCR detection (Figure 4). Still, in some cases, the growth parameters were higher in this reduced-infected control, and the disease symptoms were lesser than in the inoculated control. An example is the plants’ health and shoot and cob fresh weight on day 80 in the no-till soil (Figure 5).

Representative photos of the greenhouse experiment are shown in Figure 6. Wilt symptoms at the season’s end were typical [32,44] and included the color alteration of the leaves to light silver and then to light brown, and dehydration of the large bracts surrounding the cobs (the spathes).

### 3.2. Crop Rotation and Tillage Impact in the Field

The crop cycling and tillage system were evaluated in a commercial field having a history of mild late wilt disease [32,33]. The Megaton cv. was tested in the summer of 2017 in a nearby field (ca. 4 km away) and suffered severe dehydration and collapsed 66 DAS [64]. The aboveground sprouts’ emergence, evaluated six days post-seeding, was similar in all treatments (Table 4). However, 50 days past sowing at the end of the sprouting phase (10–11 leaves before the tasseling phase), the tillage significantly disrupted the plants’ development (number of leaves, shoot weight and height) in the control group. The clover-maize crop rotation and, to a lesser extent, the commercial Resid MG treatment were able to rescue these parameters. Yet the wheat-maize sequence was ineffective and resulted in low growth indices. Interestingly, in contrast to all other treatments, the clover soil tillage improved growth parameters (albeit statistically insignificantly).

Yield evaluation on harvest day (80 DAS, 25 DAF) revealed insignificant statistical differences between most treatments (Figure 7). The A-class (cob weight exceeding 250 g) cob yield was the lowest in the clover soil and was significantly improved in the Resid MG-treated soil (both with tillage). Nevertheless, these variations were insignificant (*p* > 0.05) in the total yield estimation.

In contrast, in the treatments’ symptoms evaluation (Figure 8), significant differences in healthy plants were identified between the clover-maize rotation, which reached the highest value, and the Resid MG treatment, which led to the lowest value (both in no-till soil). Yet, similar to the cob yield, most health values were statistically identical. The soil cultivation reduced the health values in the control and the clover soil and increased the health values in wheat and Resid MG. Finally, tracking the *M. maydis*’s DNA in the plants’ roots (50 DAS) and the above ground first internode (80 DAS) showed an increase in pathogen infection toward the season’s end (Figure 9). At 50 DAS, the tillage enhanced the pathogen present in the roots in all treatments. This effect was identified in the clover soil 80 DAS, but the tillage repressed the pathogenesis in the other treatments (statistical significance recorded only in the Resid MG-treated plants).

The field experiment’s images show the plants’ development throughout the season until the disease symptoms outburst at maturation near harvest day (Figure 10).

## 4. Discussion

Plant-fungal associations can range from mutualistic symbiosis and commensalism to parasitic, but are dependent on the host (including its rhizosphere, the research topic here) and the surroundings [67]. Such relationships can result in the plants’ improved capability to survive in stressful environments (physical and biological) [68]. An example is the soil *Trichoderma* species, known for their high myco-parasitic potential [69]. A study of the non-pathogenic microbial networks associated with maize roots could improve maize crop management and yield production under late wilt disease threat [32,34,47].

In the current study, dual-cropping experiments with or without land cultivation (tillage) were conducted simultaneously in a greenhouse and in the field to study their effect on the maize LWD. This is the first time that the hyper-susceptible hybrid, Megaton cv., was tested with this method and the first time in Israel that this practice was applied in the field. The importance of this approach has increased since the restriction of chemical fungicides has become a global priority [70]. Hence, substantial scientific efforts have been devoted to searching for alternative LWD control approaches. Many such studies focused their attention on eco-friendly substitutions for traditional chemical methods. These include late wilt green control using soil conservation practices that promote antagonizing mycorrhizal fungi (reviewed in [5,36]). While this scientific course has been broadly explored against many plant pathogens [71,72] regarding *M. maydis*, considerable information gaps exist. Subsequently, the potential of green practices to control LWD has only recently been discovered.

In our previous study carried out in Israel, we showed that preserving soil microflora completeness (by no-till practice) and manipulating its composition (by inspecting different winter crops in double-season cultivation) proved to be important [56]. In the greenhouse, wheat soil significantly improved Prelude cv. maize growth and yield while reducing LWD symptoms and the *M. maydis*’s host colonization (see the Introduction for more details). Applying tillage before the maize growing season had no significant impact on the above results.

Does the wheat and corn relationship as *Poaceae* enable them to enjoy the protection and growth promotion of shared mycorrhizal communities? In other cases, it was found that plants acquired mycorrhizal networks closely related to that of the former crop and different from that found when the soil was cultivated by tillage or left bare [57]. So, in some cases, this might be a true assumption. Yet the current study suggests a more complex picture that is highly dependent on the specific maize cultivar and growth conditions. In particular, the field environment is challenging for evaluating the benefit of such an agrotechnical approach since it is impossible to control the many variables affecting the outcome.

The outcome of these trials and the numerous data collected are complex to process and the resulting image is not unequivocal. Thus, a comparative evaluation to identify and summarize the main effects of the various treatments is needed. Therefore, we constructed a table showing the differences in percentages of each index in each treatment to the control—corn plants grown on tillage soil without pre-cultivation (Table 5).

This analysis shows that the clover-maize sequence, particularly this rotation in no-till soil, yielded the highest growth promotion and crop yield. This result was equivalent in the greenhouse and the field (Table 5). In the greenhouse, the second-best treatment was the wheat-maize rotation, a practice that was also successful in our previous study with a lesser susceptible LWD maize cultivar [56]. Yet, unlike the previous research, here the wheat soil tillage resulted in better growth. Still, the wheat soil processing enhanced the LWD symptoms in both our current and previous studies. Intriguingly, the commercial mycorrhizal (Resid MG) treatment in the field was better than the wheat-maize rotation. The high score of this treatment in the field was determined mostly because of the health index (see Figure 8). Indeed, the most highly influencing indexes in the greenhouse trial were the growth parameters (plant weight and height), whereas, in the field, the health status was the dominant parameter (see the concluding line in Table 5).

The commercial preparation (Resid MG) was more effective when tillage was applied (as in the wheat soil). This result may suggest that specific mycorrhiza species have an advantage in cultivated soils. Still, this assumption may be an oversimplification and other factors are likely to be involved. These factors may include soil microflora, oxygen enrichment during cultivation, soil density, etc. To add to the above discussion, it is essential to remember that crop cycles may have a long-term effect on soil microorganisms’ populations that may only be beneficial after a lengthy period.

Conserving and improving soil networks of mycorrhizal fungi between seasons has been vital to crop safety [60]. Indeed, arbuscular mycorrhizal fungi (AMF) belonging to the phylum *Glomeromycota* can enhance plants’ biotic and abiotic stress resistance by activating their local and systemic defense mechanisms [58]. The AMF are obligatory symbionts that inhabit 80% of terrestrial plants [72]. They are treated as “natural fertilizers” because they provide the plant with nutrients, affect root morphology and improve its water balance. These characteristics make them an important tool in modern agriculture [62]. Moreover, mycorrhiza has aggressive antagonistic relationships with many plant pathogens and is thus used as a biological pesticide [73].

Different agrotechnical approaches can build strong AMF communities in the field or disrupt them. Other crop disease studies propose that no-till cropping, crop rotation and cover crops could improve mycorrhizal communities’ establishment [57]. In contrast, intensive land cultivation and lengthy periods where the field is uncultivated may disrupt the integrity of those communities. Such soil mycorrhizal networks are also crucial in the case of maize resistance to LWD [35]. In Egypt, the influence of maize root colonization by microorganisms was also practical for decreasing the *M. maydis* impact [74]. The rhizosphere actinomycetes *Streptomyces graminofaciens*, *S. annulatus*, *S. rochei*, *S. gibsonii*, *Candida glabrata*, *C. slooffii, C. maltosa* and the fungi *Rhodotorula rubra* significantly restricted the LWD pathogen growth in vitro and in seed dressing under greenhouse conditions. Applying these species in the absence of *M. maydis* significantly improved maize plant growth indices [74].

Another research study conducted in Egypt demonstrated that rice-maize crop rotation provides some LWD control [75]. Late wilt symptoms did not develop when maize cropping was done following paddy-cultivated rice (that increases soil Mn availability) [76]. In Portugal, grain production was significantly improved, and *M. maydis* presence was substantially reduced when both cover crop and minimum tillage were applied [35]. AMF root colonization was also found to be higher following these practices.

Long-term research on the influence of different crops in rotation (including wheat–corn) and the tillage system (no-till or conventional land processing) on microbial load and other soil characteristics was conducted in southern Brazil in 1976 [77]. The no-tillage regime led to increases in microbial biomass at the 0–5-cm soil depth. Decreased tillage had a more significant impact on microbial load than crop cycling, mostly at this depth. These results prove that both tillage and crop rotation determine the microbial immobilization of soil nutrients.

It may be inferred from the current study’s results and from other studies that no sole cropping practice is ideal for all fungi [78,79]. Hence, a tailored solution may be needed to protect the crop chosen in the rotation. The tillage system and disease/pathogen stresses should also be considered. Before application on a commercial scale, such situations must be inspected and proven in short- and long-term (several years) studies. Closely related plant species such as corn and wheat may derive an advantage from a similar control strategy, but this is not an obligatory condition.

## 5. Conclusions

The current late wilt disease (LWD) common control methods have drawbacks. Previous research demonstrated that wheat-maize rotation while avoiding tillage can provide some defense against the pathogen *Magnaporthiopsis maydis*. Still, this preliminary work left many unanswered questions, particularly about the effect of this approach in other maize hybrids and under commercial field conditions. To meet this challenge, the current study applied these agrotechnical practices in parallel as dual-crop experiments—in the greenhouse and in the field—using a hyper-susceptible maize hybrid (Megaton cv.). The clover-maize rotation, especially in the no-till soil, was superior to the other practices tested (the wheat-maize cycle and the commercial Resid MG preparation) and was reflected in growth promotion and LWD durability. The wheat-maize sequence was successful in the greenhouse but ineffective in the field. Surprisingly, the commercial Resid MG soil treatment, which had no evident advantage in the greenhouse, resulted in impressive plant health promotion in the tillage plots in the field. To conclude from the current work and other studies, choosing the specific crop cycle, cover crop and tillage system could reduce LWD pressure and assist in restricting fungicide use, which has adverse risks to the environment and human health. Since different maize cultivars may react differently to such agrotechnical practices, each cultivar should be tested separately to determine its specific LWD protection suite. Field studies and long-term studies are needed to fully understand the potential of this approach.

## Figures and Tables

**Figure 1 jof-08-00586-f001:**
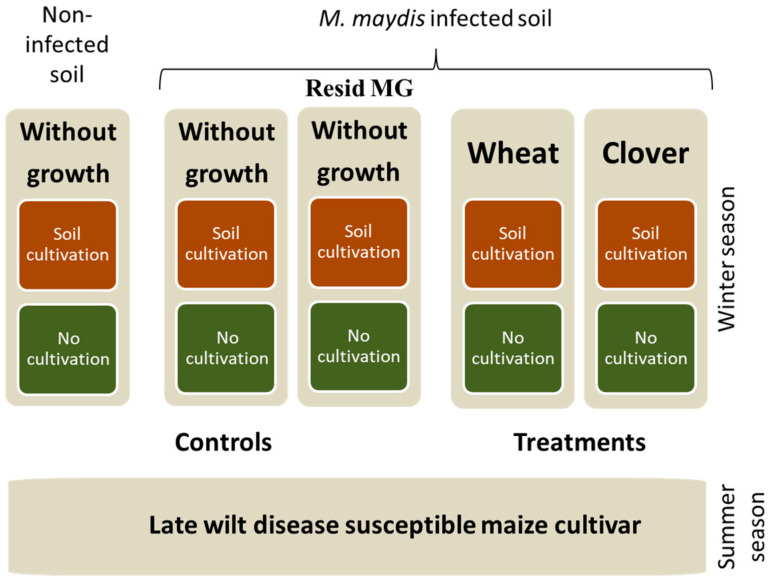
The greenhouse experiment program. The experiment was performed in greenhouse pots over an entire dual-growing season (winter cultivation followed by spring-summer cultivation). The late wilt disease (LWD) hyper-susceptible maize genotype Megaton cv. was grown on *Magnaporthiopsis maydis*-infected soil and treated as follows: clover/maize crop rotation and wheat/maize crop rotation. The controls included the commercial Resid MG product (arbuscular mycorrhiza), which was tested as a soil addition instead of winter cropping. This product is based on *Glomus iranicum* var. *tenuihypharum* (Mico, Symborg S.L., Murcia, Spain, supplied by BioBee Biological Systems, Sde Eliyahu, Israel) and was added to the bare soil one week post maize sowing. Infected and uninfected soils that had not been cropped before the maize growth were also used as reference treatments.

**Figure 2 jof-08-00586-f002:**
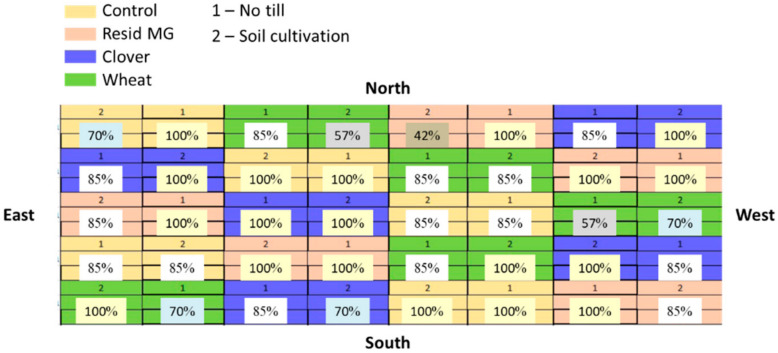
The field experiment map and maize emergence percentages. An LWD-contaminated field (Gadash Experimental Farm, Upper Galilee, Hula Valley, Northern Israel) was used to evaluate the winter crop (clover or wheat) and tillage system on disease severity and pathogen population in the hyper-susceptible Megaton cv. The treatments were compared to a commercial Resid MG treatment (arbuscular mycorrhiza soil enrichment) and non-cropped soil (the soil left bare throughout the winter). A split-plot design was applied, whereby (1) indicates no-till cropping and (2) traditional soil cultivation (tillage). Each treatment/control was repeated five times and the plots were scattered in the field using a completely randomized design. Above ground sprouts’ emergence percentages were stated six days after sowing (DAS) and indicated for each plot.

**Figure 3 jof-08-00586-f003:**
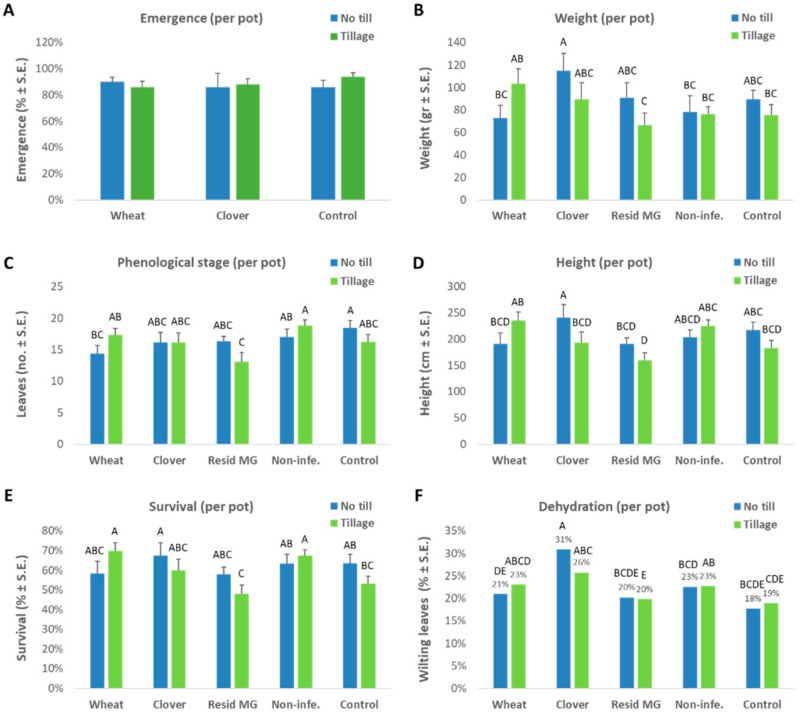
The greenhouse pot experiment—emergence evaluation (3 DAS) and growth parameters at the end of the sprouting phase (45 DAS). The greenhouse trial is described in Figure 1. The plants’ growth parameters (average per pot) included aboveground surface appearance (**A**), shoot fresh weight (**B**), phenological stage (the number of leaves, **C**), and height (**D**). The Resid MG (or Mico) product powder was added one week after the sowing, and thus this treatment was not included in the emergence evaluation. The plants’ health status at that age was evaluated using survival percentages (**E**) and wilt assessment (**F**) that were based on the percentage of dry leaves in each repeat. Non-infe.—reduced-infected soil control. Values represent an average of 10 replications (pots, each containing five plants) ± standard error. Statistically significant differences (one-way ANOVA, *p* < 0.05) between treatments at the same measures (if they exist) are indicated by different letters (A–E).

**Figure 4 jof-08-00586-f004:**
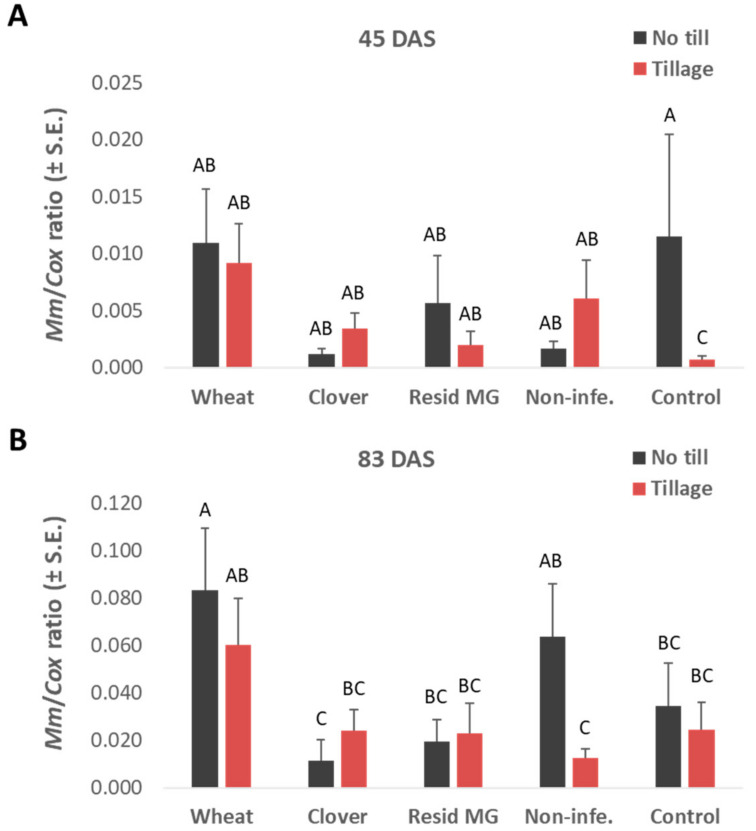
Greenhouse real-time PCR (qPCR) analysis measures of the pathogen DNA at 45 (**A**) and 83 (**B**) DAS in the plants’ roots and stems’ aboveground first internode, respectively. The experiment is described in Figure 1. Non-infe.—reduced-infected soil control. The *Y*-axis represents the amount of DNA of the *M. maydis* pathogen relative to the total DNA in the plant tissue (represented by cytochrome c oxidase (COX) gene DNA present in plant tissue and the fungus cells). The values represent an average of 10 biological repetitions and error lines represent a standard error. Statistically significant differences (one-way ANOVA, *p* < 0.05) between treatments at the same measures are indicated by different letters (A–C).

**Figure 5 jof-08-00586-f005:**
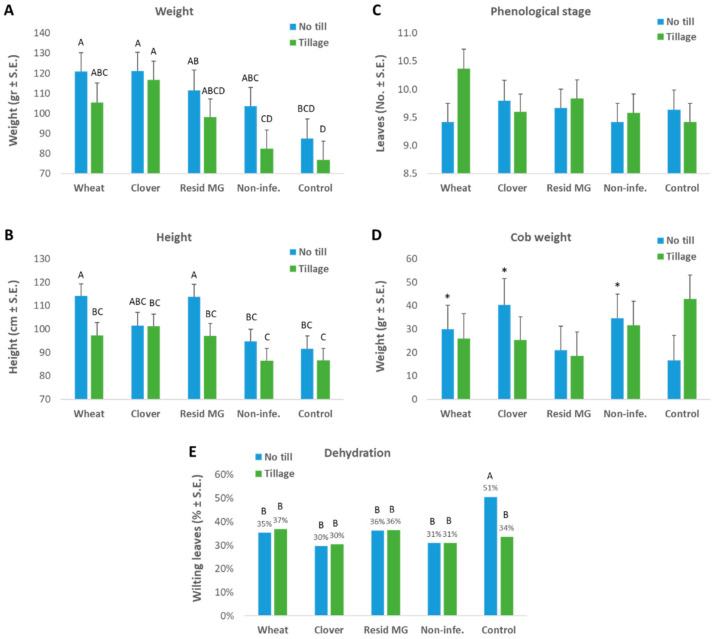
The greenhouse pot experiment—growth parameters at 83 DAS (28 days after fertilization, DAF). The experiment is described in Figure 1. The plants’ growth parameters included fresh weight (**A**), plant height (**B**), phenological stage (**C**), cob yield (**D**) and the plants’ health status (**E**). Non-infe.—reduced-infected soil control. Wilt assessment was based on the percentage of dry leaves in each repeat. Values represent an average of 10 biological replications (pots, each containing one plant) ± standard error. Statistically significant differences (one-way ANOVA, *p* < 0.05) between treatments at the same measures (if they exist) are indicated by different letters (A–D). Asterisks represent a significant difference (*t*-test, *p* < 0.05) from the tillage soil at the same treatment.

**Figure 6 jof-08-00586-f006:**
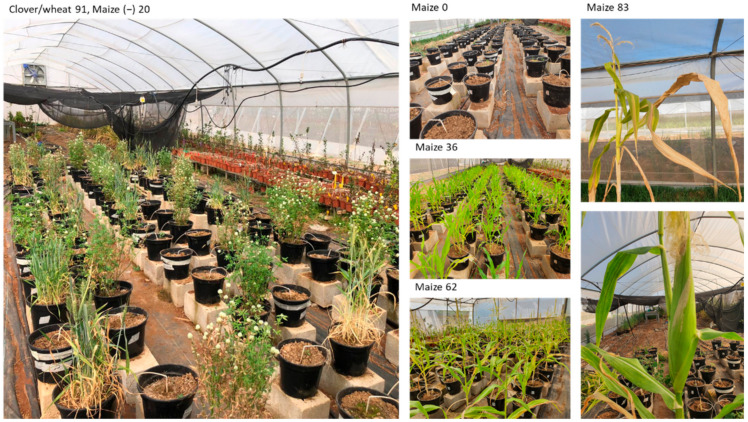
The greenhouse pot experiment—representative photos. The experiment is described in Figure 1. Numbers indicate days from sowing.

**Figure 7 jof-08-00586-f007:**
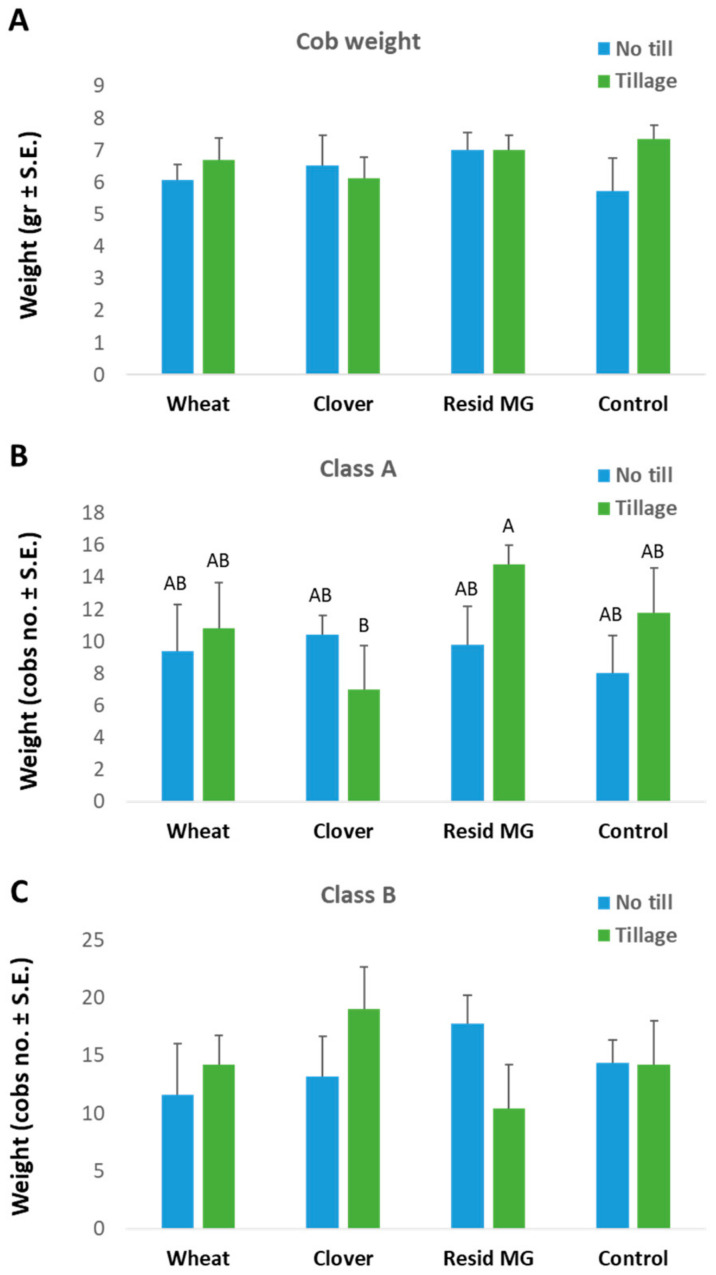
Yield assessment of the field experiments. The evaluation was carried out 80 DAS (25 DAF) and included the total cobs’ average weight in each treatment (**A**), A-class yield—the number of cobs with a weight exceeding 250 g (**B**), and B-class yield— the number of cobs with a weight below 250 g (**C**). Values indicate an average of five replications (each experiment repetition sample included all cobs in the area (15-37 cobs per repeat), so a total of 105-138 cobs were collected for each treatment). Error bars indicate standard error. Statistically significant differences (one-way ANOVA, *p* < 0.05) between treatments at the same measures (if they exist) are indicated by different letters (A, B).

**Figure 8 jof-08-00586-f008:**
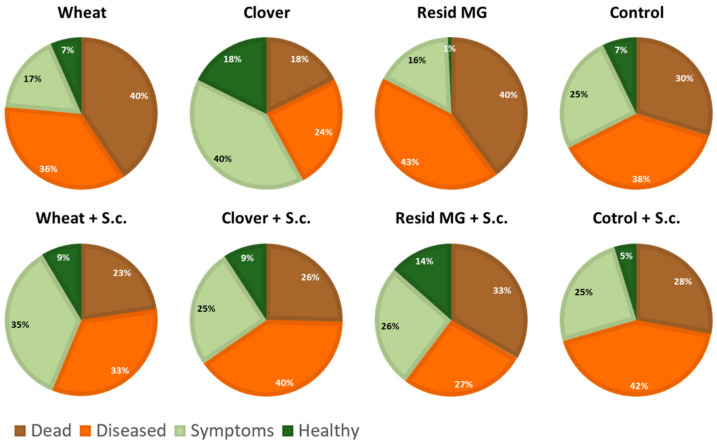
Field wilt assessment. Values were determined at 80 DAS (25 DAF). The experiment is described in Figure 2. Statistically significant differences (one-way ANOVA, *p* < 0.05) were found in the healthy plants between the clover and the Resid MG treatments, in the symptomatic plants between the clover and the wheat treatments and in the dead plants between the clover and the Resid MG treatments (all without tillage).

**Figure 9 jof-08-00586-f009:**
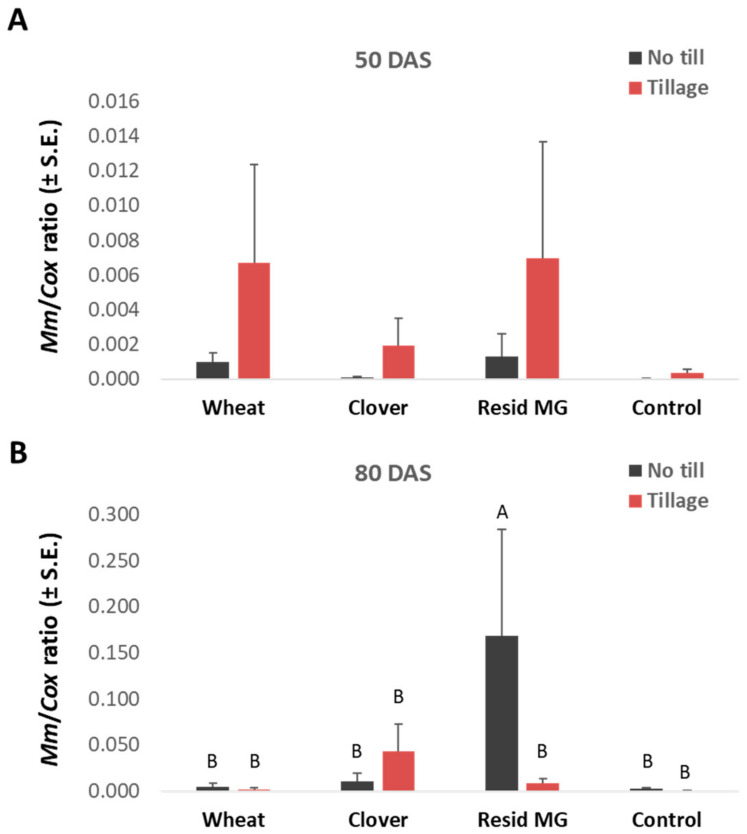
Field real-time PCR (qPCR) analysis measures of the pathogen DNA at 50 (**A**) and 80 (**B**) DAS in the plants’ roots and aboveground first internode, respectively. The experiment is described in Figure 2. The *Y*-axis represents the amount of DNA of the *M. maydis* pathogen relative to the total DNA in the plant tissue (represented by cytochrome c oxidase (COX) gene DNA present in the plant tissue and fungus cells. The values represent an average of five biological repetitions and error lines represent a standard error. Statistically significant differences (one-way ANOVA, *p* < 0.05) between treatments at the same measures (if they exist) are indicated by different letters (A, B).

**Figure 10 jof-08-00586-f010:**
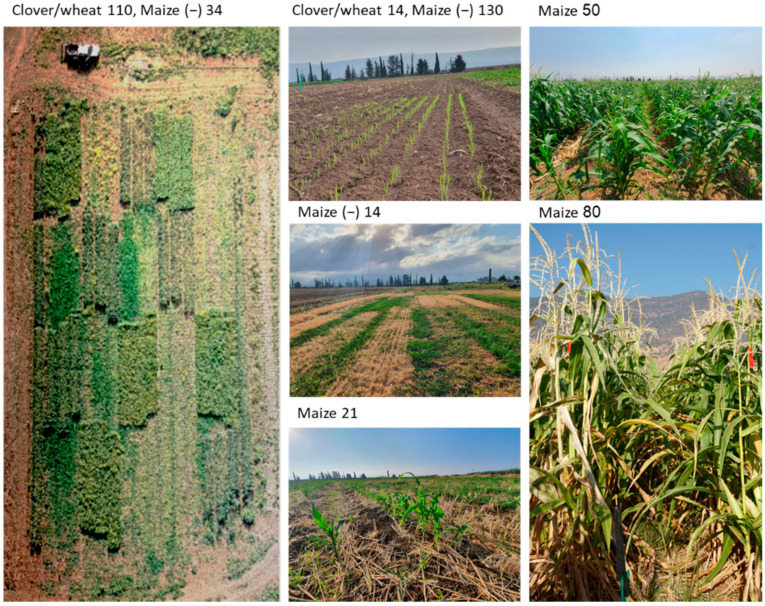
The field experiment—representative photos from the air (on the **left**) and the ground (on the **right**). The field trial is described in Figure 2. Numbers indicate days from sowing.

**Table 1 jof-08-00586-t001:** Timetable for the greenhouse and field experiments.

Greenhouse Experiment			
Procedure	Date	Days from Sowing	Remarks
Soil inoculation	23/11/2020		22 days before winter sowing
Clover and wheat sowing	15/12/2020	0	
Clover and wheat harvest	16/03/2021	91	
Land tillage	01/04/2021	-	5 days before maize season
Maize sowing	05/04/2021	0	20 days post-winter season
Emergence evaluation	08/04/2021	3	
Commercial mycorrhiza treatment	12/04/2021	7	Resid MG prepartation
Thining and sprout phase evaluation	23/05/2021	45	
Fertilization	30/05/2021	55	Female flowering day
Maize harvest	27/06/2021	83	28 days after fertilization
**Field Experiment**			
**Procedure**	**Date**	**Days from Sowing**	**Remarks**
Clover and wheat sowing	13/01/2021	0	
Clover and wheat harvest	17/05/2021	124	
Land tillage	27/05/2021	-	10 days before maize season
Maize sowing	06/06/2021	0	20 days post-winter season
Commercial mycorrhiza treatment	06/06/2021	0	Resid MG prepartation
Emergence evaluation	12/06/2021	6	
Sprout phase evaluation	26/07/2021	50	
Fertilization	31/07/2021	55	Female flowering day
Maize harvest	25/08/2021	80	25 days after fertilization

**Table 2 jof-08-00586-t002:** Meteorological data for the 2021 dual season experiment ^1^.

Parameters	Winter Season (Clover/Wheat)	Summer Season (Maize)
Dates	January 13–May 17	June 6–August 25
Temperature (°C)	15.6 ± 7.3	27.6 ± 5.3
Humidity (%)	70.4 ± 23.3	60.0 ± 18.5
Soil temp. top 5 cm (°C)	17.0 ± 5.6	30.4 ± 3.2
Radiation (W/m^2^)	184.5	296.9
Precipitation (mm)	395.6	0
Evaporation (mm)	515.5	672.1

^1^ Data (average ± standard deviation) according to Israel Northern Research and Development (Hava 1 Meteorological Station) data.

**Table 3 jof-08-00586-t003:** Primers for *Magnaporthiopsis maydis* qPCR detection ^1^.

Pairs	Primer	Sequence	Uses	Amplification	References
Pair 1	A200a-for A200a-rev	5′-CCGACGCCTAAAATACAGGA-3′ 5′-GGGCTTTTTAGGGCCTTTTT-3′	Target gene	200 bp *M. maydis* species-specific fragment	[16]
Pair 3	COX-F COX-R	5′-GTATGCCACGTCGCATTCCAGA-3′ 5′-CAACTACGGATATATAAGRRCCRR AACTG-3′	Control and normalization	Cytochrome C oxidase (*COX*) gene product	[65,66]

^1^ The real-time PCR (qPCR) reactions were performed using the ABI PRISM 7900 HT Sequence Detection System (Applied Biosystems, Waltham, MA, USA) and 384-well plates. The 5-µL qPCR reaction mixture was applied to a well comprised of 2 µL of DNA sample extract, 2.5 µL of iTaq™ Universal SYBR Green Supermix (Bio-Rad Laboratories Ltd., Hercules, CA, USA), and 0.25 µL (10 µM each) of the forward and reverse primers. The qPCR cycle program was activation phase (pre-cycle), 1 min at 95 °C, 40 denaturation cycles (15 s at 95 °C), annealing and extension (30 s at 60 °C) and finalizing phase by a melting curve analysis.

**Table 4 jof-08-00586-t004:** The field experiment—growth parameters at 50 DAS ^1^.

	Soil Cultivation	Emergence (%) 6 DAS		Shoot Wet Weight (g)		Leaves (no.)		Height (cm)	
**Control**	-	94% ± 4%	AB	97 ± 2.5	A	11.4 ± 0.05	A	154 ± 1.0	AB
	+	88% ± 6%	AB	73 ± 1.3	C	10.6 ± 0.04	C	148 ± 0.2	BC
**Resid MG**	-	100% ± 0%	A	84 ± 1.2	ABC	10.6 ± 0.07	C	156 ± 0.8	A
	+	82% ± 11%	AB	80 ± 1.0	BC	10.4 ± 0.07	C	145 ± 0.9	C
**Clover**	-	88% ± 3%	AB	84 ± 1.4	ABC	10.8 ± 0.05	BC	154 ± 0.5	AB
	+	94% ± 6%	AB	87 ± 2.1	AB	11.2 ± 0.07	AB	160 ± 0.6	A
**Wheat**	-	76% ± 6%	B	73 ± 0.9	C	10.4 ± 0.08	C	145 ± 0.7	C
	+	82% ± 8%	AB	74 ± 0.9	C	10.3 ± 0.04	C	144 ± 0.5	C

^1^ Emergence percentages were measured six days after sowing. All other data were collected after 50 days of growing maize sprouts (Megaton cv.). Values indicate an average of five replications ± standard error (each experiment repetition sample included five plants, so a total of 25 plants was collected for each treatment). Statistically significant differences (one-way ANOVA, *p* < 0.05) between treatments at the same measures are indicated by different letters (A–C) at the right of each column.

**Table 5 jof-08-00586-t005:** Comparative analysis of the greenhouse and field experiments results ^1^.

Treatment		Pot Experiment										
Crop Rotation	Land Processing	Eme. (%)	Weight (g)		Leaves (no.)		Height (cm)		Surv. (%)	Hea. (%)	Total (av.)	Total (sum)
		3 DAS	45 DAS	83 DAS	45 DAS	83 DAS	45 DAS	83 DAS	45 DAS	83 DAS		
**Control**	*No till*	−9	18	14	14	2	19	6	19	−50	**4**	**4**
**Resid-MG** **control**	*No till*		20	45	0	3	5	31	9	−8	**13**	**11**
	*Soil cultivation*		−12	28	−19	4	−13	12	−10	−8	**−2**	
**Clover**	*No till*	−9	52	58	0	4	32	17	26	12	**21**	**34**
	*Soil cultivation*	−6	18	52	−1	2	6	17	13	9	**12**	
**Wheat**	*No till*	−4	−4	57	−12	0	5	32	9	−5	**9**	**25**
	*Soil cultivation*	−9	36	37	7	10	29	12	31	−10	**16**	
**Total (av.)**		**−7**	**18**	**42**	**−2**	**4**	**12**	**18**	**14**	**−9**		
**Treatment**		**Field experiment**										
**Crop Rotation**	**Land Processing**	**Eme. ** **(%)**	**Weight** **(g)**		**Leaves** **(no.)**		**Height** **(cm)**		**Cobs** **(g)**	**Hea. ** **(%)**	**Total** **(av.)**	**Total** **(sum)**
		**6** **DAS**	**50** **DAS**		**50** **DAS**		**50** **DAS**		**80** **DAS**	**80** **DAS**		
**Control**	*No till*	7	32		7		4		−22	33	**10**	**10**
**Resid-MG** **control**	*No till*	14	15		0		5		−4	−83	**−9**	**21**
	*Soil cultivation*	−6	9		−2		−2		−4	183	**30**	
**Clover**	*No till*	0	14		1		4		−11	250	**43**	**64**
	*Soil cultivation*	7	19		6		8		−17	100	**21**	
**Wheat**	*No till*	−13	0		−3		−3		−17	17	**−3**	**7**
	*Soil cultivation*	−6	0		−3		−3		−9	83	**10**	
**Total (av.)**		**0**	**13**		**1**		**2**		**−12**	**83**		

^1^ The plants’ aboveground emergence (Eme.), growth, yield, survival (Surv.) and health (hea.) results throughout the season were analyzed by calculating the differences in percentages of each index in each treatment to the control—corn plants that were grown on tillage soil without pre-cultivation. The total column or row (highlighted in bold fonts) is the average of all score measures. The right-most column summarizes the total sum of each row. The data highlighted in orange were significantly different (*p* < 0.05) from the control (ANOVA with *t*-test post hoc).

## Data Availability

The datasets generated during and/or analyzed during the current study are available from the corresponding author on reasonable request.

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
