# Peer review of "Crop Rotation and Minimal Tillage Selectively Affect Maize Growth Promotion under Late Wilt Disease Stress"

_jof, 2022, doi:10.3390/jof8060586_

Round 1

Reviewer 1 Report

The investigated how authors on how tillage and affect maize growth under  Maize late wilt disease stress. Overall the paper is well written and scientifically important and relevant. The manuscript can therefore be accepted in its current form with minor changes (see below).

  1. The authors need to explain in more detail aims and objectives of the study. For example, in the last paragraph of the Introduction they refer using "this method". If they can explain in words what the method is what they used (Lines 114 and 117).
  2. In Figure 2, the colours are going over the lines. If the authors can correct this.
  3. The resolution of Figure 2 can also be improved.

Author Response

Responses to Reviewer 1’s comments

We thank the reviewer for investing substantial efforts, which are undoubtedly contributing to this manuscript. The remarks and suggestions improved this paper’s scientific soundness and accurateness. Your contribution is greatly appreciated.

The investigation on how tillage and affect maize growth under maize late wilt disease stress. Overall the paper is well written and scientifically important, and relevant. The manuscript can therefore be accepted in its current form with minor changes (see below).

Reply: Thank you for the positive evaluation of our manuscript. All your remarks and suggestions were carefully and thoroughly addressed, as detailed below.

  1. The authors need to explain in more detail the aims and objectives of the study. For example, in the last paragraph of the Introduction, they refer to using “this method”. If they can explain in words what the method is what they used (Lines 114 and 117).

Reply: The reviewer is correct. This sentence was rewritten and now reads: “The above findings encourage further exploration of the crop rotation (clover/wheat) with maize and the effect of tilling on the impact of LWD on maize development and yield.” (Lines 115-117)

  1. In Figure 2, the colors are going over the lines. If the authors can correct this.
  2. The resolution of Figure 2 can also be improved.

Reply: We improved Figure 2. We hope that this answers your concerns.

Reviewer 2 Report

The authors investigated the effect of crop rotation (wheat/clover) with maize and the effect of tilling on the impact of LWD on maize development and yield.

The paper is very well written and easy to follow. The introduction gives a good introduction into the subject.

The general experimental idea is very smart: using both greenhouse trials (under controlled conditions and with all useful controls that cannot be applied in the open field easily) and field trials (needed to really get an idea on the performance under practical conditions). But still to me the paper has major issues that need to be solved before publication.

My major issues:

  1. The results of the qPCR: Fig. 4 (and slightly in Fig. 9): why do you have a signal in the control? Even if some contamination from neighbouring pots has happened it does not explain the high abundance in Fig. 4, even higher than many inoculated treatments. In my opinion the results are worthless like this if you have such high signals in the control.
  2. The discussion until line 556 is no discussion at all (and also in the following part to a lower extend), it has to focus much more on the actual results and not on the subject itself.
  3. Further I do not think that the siginificance of the results which are in my opinion unambiguous are justified for the impact factor of Journal of Fungi. The yield in the field trials was not significantly affected.

Minor issues:

In the acstract:

Explain LWD when first mentioned, it is explained later but should be when first mentioned.

 Line 120: do you mean artificially infected? Otherwise it is not clear what the difference is to contaminated (line 123).

Line 160: and incubated is twice.

Lines 173-180: they are acutally more presenting what has to be in the figure legend of Fig.1 than the actual legend. The legend itself is more a description of the trial. The abbreviation is missing in the legend.

Line 207: how many seeds? It is explained later, but please mention here.

Firg. 2: It has to be Resid MG, not Rasid MG.

Chapter 2.4: you mention two papers (65 and 66) as origin for the COX-primers, but the papers do not deal with maize. How did you test their specificity?

Fig. 3 legend: I do not understand how you transferred the 4 categories of wilting into the percentages you show in the y-axis. This is not clear.

General in the results: refer more to the figures (Fig 3a, b, c….) in the main text, it makes it much easier to follow your descriptions.

Line 394: you say here 40 and 80 DAS, in the figure you have 45 and 83 DAS. What is the correct number?

Fig. 7: since there is no significant difference it might be useful to skip these data in parts and add other parts to the supplements.

Fig. 8: To me the presentation of the data is not logical. It does not really make sense to me.

I would add table 5 to the main results.

Author Response

Responses to Reviewer 2’s comments

We want to express our sincere appreciation to the reviewer for the essential and helpful advice. The time and effort invested are greatly appreciated and certainly contributed to the manuscript and improved it. Thank you.

The authors investigated the effect of crop rotation (wheat/clover) with maize and the effect of tilling on the impact of LWD on maize development and yield.

The paper is very well written and easy to follow. The introduction gives a good introduction into the subject.

The general experimental idea is very smart: using both greenhouse trials (under controlled conditions and with all useful controls that cannot be applied in the open field easily) and field trials (needed to really get an idea on the performance under practical conditions). But still to me the paper has major issues that need to be solved before publication.

Reply: Thank you for the positive evaluation of our manuscript. All your remarks and suggestions were carefully and thoroughly addressed, as detailed below.

My major issues:

  1. The results of the qPCR: Fig. 4 (and slightly in Fig. 9): why do you have a signal in the control? Even if some contamination from neighbouring pots has happened it does not explain the high abundance in Fig. 4, even higher than many inoculated treatments. In my opinion the results are worthless like this if you have such high signals in the control.

Reply: Thank you for this important remark. As explained in lines 196-199: “For the negative control, local peat soil was collected from the experimental farm’s field. This control soil is similar to the infected soil collected ca. 4 km nearby. It has no known history of LWD infestation (if such infestation exists, it is estimated to be minor).”

This was made to achieve as similar as possible control soil conditions to the infected treatments’ soil. Yet, despite our expectations, the soil that had not undergone deliberate inaction had some contamination with the pathogen reflected in the highly sensitive qPCR detection. Still, in some cases, the growth parameters were higher in this reduced-infected control, and the disease symptoms were lesser than in the inoculated control. An example is the plants’ health and shoot and cob fresh weight on day 80 in the no-till soil.

The following explanation was added to the text (lines 437-445): “As explained in the Materials and Methods section, for the negative control, local peat soil was collected from the experimental farm’s fields, which had no known history of LWD infestation (if such infestation exists, it is estimated to be minor). This control aimed at achieving as similar as possible soil conditions to the infected treatments’ soil. Yet, despite our expectations, the soil that had not undergone deliberate inaction had some contamination with the pathogen reflected in the highly sensitive qPCR detection (Figure 4). Still, in some cases, the growth parameters were higher in this reduced-infected control, and the disease symptoms were lesser than in the inoculated control. An example is the plants’ health and shoot and cob fresh weight on day 80 in the no-till soil (Figure 5).”

  1. The discussion until line 556 is no discussion at all (and also in the following part to a lower extend), it has to focus much more on the actual results and not on the subject itself.

Reply: As you suggested, the Discussion section was reorganized and updated, and is now focused more on the results.

  1. Further I do not think that the siginificance of the results which are in my opinion unambiguous are justified for the impact factor of Journal of Fungi. The yield in the field trials was not significantly affected.

Reply: We understand your concern; it is challenging to achieve significant statistical differences in such experiments. The crop rotation and tillage system (even if applied in the field for several years to build a strong mycorrhizal network defense) probably won’t be sufficient to prevent LWD in highly infected areas planted with susceptible maize cultivars. Nevertheless, such an agrotechnical approach can reduce the disease’s damage and, if properly combined with other methods (such as optimal irrigation and biological pesticides), could be important for commercial production. The other reviewer of this manuscript found this work to be “scientifically important and relevant.” We believe the findings are an essential contribution to LWD research, which is receiving little attention due to the limited global distribution of the disease.

Minor issues:

In the abstract: Explain LWD when first mentioned; it is explained later but should be when first mentioned.

Reply: The abbreviation LWD is described in the opening sentence of the Abstract (Lines 10-12): “In recent years, worldwide scientific efforts towards controlling maize late wilt disease (LWD) have focused on eco-friendly approaches that minimize the environmental impact and health risks.”

Line 120: do you mean artificially infected? Otherwise, it is not clear what the difference is to contaminated (line 123).

Reply: Yes, artificially infected. The sentence was corrected to explain this better: “Crop rotations of clover/maize or wheat/maize and tillage regimes (no-tillage or conventional tillage) were examined in greenhouse pots and in the field under LWD stress using the hyper-susceptible maize cultivar, Megaton cv.” (Lines 120-122)

Line 160: and incubated is twice.

Reply: Corrected (the duplication was deleted).

Lines 173-180: they are actually more presenting what has to be in the figure legend of Fig.1 than the actual legend. The legend itself is more a description of the trial. The abbreviation is missing in the legend.

Reply: As suggested, the paragraph and Figure 1 legend were rewritten.

The paragraph now reads: “The greenhouse experiment was conducted at the R&D North Israel Experimental Farm located in the Upper Galilee, Hula Valley, northern Israel (33°09’08.2 “N 35°37’21.6 “E). This study included two treatments, commercial control, positive control, and negative control. The treatments included crop cycle with clover or wheat (planted before maize) in dual cultivation with or without tillage, as described in Figure 1. Controls without winter cropping included the addition of commercial mycorrhiza preparation, infected control (naturally infested soil with the addition of complementary M. maydis infection), and non-infected soil.” (Lines 173-180)

The Figure 1 legend now reads: “The greenhouse experiment program. The experiment was performed in greenhouse pots over an entire dual-growing season (winter cultivation followed by spring-summer cultivation). The late wilt hyper-susceptible maize genotype Megaton cv. was grown on Magnaporthiopsis maydis-infected soil and treated as follows: clover/maize crop rotation and wheat/maize crop rotation. The controls included the commercial Resid MG product (arbuscular mycorrhiza), which was tested as a soil addition instead of winter cropping. This product is based on Glomus iranicum var. tenuihypharum (Mico, Symborg S.L., Murcia, Spain, supplied by BioBee Biological Systems, Sde Eliyahu, Israel) and was added to the bare soil one week post maize sowing. Also, infected and uninfected soils that had not been cropped before the maize growth were used as reference treatments.”

We couldn’t find any missing abbreviation in the Figure 1 legend.

Line 207: how many seeds? It is explained later, but please mention here.

Reply: This explanation was added to the text (line 207): “Each pot was sown with five seeds.”

Fig. 2: It has to be Resid MG, not Rasid MG.

Reply: We apologize for this mistake. The term “Rasid MG” was corrected to “Resid MG” throughout the text and in the figures, as advised.

Chapter 2.4: you mention two papers (65 and 66) as origin for the COX-primers, but the papers do not deal with maize. How did you test their specificity?

Reply: Indeed, but all eukaryotic organisms share this gene. Cytochrome c oxidase catalyzes the transfer of electrons from cytochrome c to oxygen during the final step of the respiratory chain (reviewed by Capaldi et al. 1983). The protein is ubiquitous to all aerobic cells. In eukaryotes, the enzyme is localized to the mitochondrial inner membrane. Our samples include plant and fungal DAN, which both have the cox gene. This reference gene is stable and easily identified and amplified.

Fungal mitochondrial genomes usually harbor 14 core genes encoding proteins involved in electron transport and oxidative phosphorylation, including the cytochrome c oxidase (cox1, cox2, cox3). See, for example, the following reference: Franco MEE, López SMY, Medina R, Lucentini CG, Troncozo MI, et al. (2017), The mitochondrial genome of the plant-pathogenic fungus Stemphylium lycopersici uncovers a dynamic structure due to repetitive and mobile elements. PloS One 12(10): e0185545. https://doi.org/10.1371/journal.pone.0185545

The specific M. maydis qPCR detection was validated, approved and published (Degani et al., Plant Disease, 2019, 103, 238-248). We have used this qPCR method repeatedly in several additional studies (see, for example, Degani et al., PloS One 2018, 13, e0208353; Degani et al., Agronomy 2019, 9, 181; Dor and Degani, Plants 2019, 8; Degani et al., Microorganisms 2020, 8). The same protocol with some adjustments is used worldwide in the scientific community for a similar purpose (identifying pathogens’ DNA inside host tissues).

Fig. 3 legend: I do not understand how you transferred the 4 categories of wilting into the percentages you show on the y-axis. This is not clear.

Reply: Thank you for this remark; this issue needs to be explained more clearly. The four categories of wilting were only used in the field experiment. In the greenhouse experiment, we counted the number of dry leaves out of the total number of leaves and calculated their percentages, as detailed below.

In the greenhouse experiment:

  • On day 45, we sampled ca. 130-210 leaves from each treatment/control (a total of 1,855 leaves were collected). The number of dry leaves was stated in each repeat, and the percentage of those leaves was calculated.
  • On day 83, we sampled 100-120 leaves from each treatment/control (a total of 1,121 leaves were collected). The number of dry leaves was stated in each repeat, and the percentage of those leaves was calculated.

In the field experiment, the sampling was made of the whole plant. We sampled 110-140 plants from each treatment (a total of 987 plants were evaluated). The plants were categorized according to their LWD symptoms. The average plants’ numbers in each disease stage in each treatment were calculated and presented.

The above explanation was added to Materials and Methods (Section 2.2.4., lines 233-238, and Section 2.3.2., lines 305-307).

General in the results: refer more to the figures (Fig 3a, b, c….) in the main text, it makes it much easier to follow your descriptions.

Reply: We agree. The reference to the figures’ subsections was added to the description of Figures 3 and 5, and Table 5.

Line 394: you say here 40 and 80 DAS, in the figure you have 45 and 83 DAS. What is the correct number?

Reply: The sampling days were 45 and 83 for the midperiod and season-end evaluations in the greenhouse experiment, respectively. In the field experiment, the sampling days were 50 and 80, respectively. We double-checked that all of the dates are correct in the text and figures and corrected this in several places.

Fig. 7: since there is no significant difference, it might be useful to skip these data in parts and add other parts to the supplements.

Reply: There are statistical differences in the A-class cob weight. We updated this figure and added the missing information (we apologize for unintentionally not including this information in the first version). Indeed, no statistical differences can be identified in the other two figures. This is very common in field experiments due to the high variability in the environmental (atmospheric and soil) conditions. It is also the result of the uniform disease spreading in the field (see Degani, J. Fungi 20217(11), 989; https://doi.org/10.3390/jof7110989). Thus, the high standard error of the data makes it challenging to receive significance at α < 0.05. Yet, prominent differences do exist (ca. 10-20%, see Table 5), and we believe it is important to present them in the main body text.

Fig. 8: To me the presentation of the data is not logical. It does not really make sense to me.

Reply: The Figure 8 (field wilt assessment) data were rearranged and are now presented as a pie chart in the same order as the other figures (wheat, clover, Resid MG and control). The figure legend was rewritten. We hope that this answers your concern.

I would add table 5 to the main results.

Reply: Table 5 is a comparative description of the results and is an integral part of the Discussion. Thus, we think it would be best to present it near the text that refers to it. Yet, this is a minor issue that could be easily resolved. We leave this decision to the Editor.

Reviewer 3 Report

please see attach file  

Author Response

Responses to Reviewer 3’s comments

We would like to express our sincere appreciation to the reviewer for the essential and helpful advice. The time and effort invested are greatly appreciated and certainly contributed to the manuscript and improved it. Thank you.

Page: 1

L10-17 - The abstract did not mention any number on how it reduction of disease

Reply: This is correct. We added the missing information to the Abstract and the paragraph now reads: “In the greenhouse under LWD stress, the results clearly demonstrate the beneficial effect of maize crop rotation with clover and wheat on plant weight (1.4-fold), height (1.1-1.2-fold) and cob yield (1.8-2.4-fold), especially in the no-till soil. The clover-maize growth sequence excels in reducing disease impact (1.7-fold) and pathogen spread in the host tissues (3-fold). Even though the wheat-maize crop cycle was less effective, it still had better results than the commercial mycorrhizal preparation treatment and the uncultivated non-infected soil. The results were slightly different in the field. The clover-maize rotation also achieved the best growth promotion and disease restraint results (2.6-fold increase in healthy plants), but the maize rotation with wheat had only minor efficiency.” (lines 21-29)

Page: 2

L65-80 - I think you can omit this part

Reply: Since the main focus of this work is to test a new eco-friendly and financially feasible approach to restrain late wilt disease damage, it is essential to describe the accumulated knowledge on the topic and the current control methods that were evaluated and used. Besides the importance of the data included in this paragraph, in our opinion, the text clarity and the logic presented in the Introduction would lose their focus if we omit this section.

Page: 3

L114-122 - The aim like abstract, not objective it needs to rewrite

Reply: The paragraph was rephrased and is now reads: “This method was examined thoroughly over double cultivation and two seasons of study in the current work. Crop rotations of clover/maize or wheat/maize and tillage regimes (no-tillage or conventional tillage) were examined in greenhouse pots and in the field under LWD stress using the hyper-susceptible maize cultivar, Megaton cv. These treatments were compared to a commercial mycorrhizal preparation (Resid MG [62], Mico, BioBee, Sde Eliyahu, Israel), and infected and non-infected bare soil controls. The efficiency of these agrotechnical practices on M. maydis pathogenesis was evaluated by following the plants’ growth and yield parameters, estimating disease severity, and quantifying the pathogen’s DNA inside the plants’ tissues using a real-time (qPCR)-based technique.” (lines 118-127)

Page: 4

L157-158 - Is it important to use the same amount of fungi?

Reply: Yes. This will enable other researchers to repeat this procedure and achieve comparable results.

L160 - delete it: “and incubated”

Reply: This was corrected (the duplication was deleted).

Page: 5

L194 - correct it: “1.6 x 104”

Reply: corrected to 1.6 x 104.

Page: 13 - Figure 6 - you can delete it

Page: 16 - Figure 10 - I do not think this is necessary

Reply: The first and second reviewers did not recommend this (deleting Figures 6 and 10). We agree with them. We believe that the experiment photos and the disease symptoms provide essential information that should be presented. We will leave this decision to the Editor.

Page: 19

Conclusions, L622-646 - it is too long and needs to shortage

Reply: We agree and corrected (shortened) the Conclusion section, which now reads: “The current late wilt disease (LWD) common control methods have drawbacks. Previous research demonstrated that wheat-maize rotation while avoiding tillage can provide some defense against the pathogen Magnaporthiopsis maydis. Still, this preliminary work left many unanswered questions, particularly about the effect of this approach in other maize hybrids and under commercial field conditions. To meet this challenge, the current study applied these agrotechnical practices in parallel as dual-crop experiments – in the greenhouse and in the field –  using a hyper-susceptible maize hybrid (Megaton cv.). The clover-maize rotation, especially in the no-till soil, was superior to the other practices tested (the wheat-maize cycle and the commercial Resid MG preparation), and was reflected in growth promotion and LWD durability. The wheat-maize sequence was successful in the greenhouse but ineffective in the field. Surprisingly, the commercial Resid MG soil treatment, which had no evident advantage in the greenhouse, resulted in impressive plant health promotion in the tillage plots in the field. To conclude from the current work and other studies, choosing the specific crop cycle, cover crop and tillage system could reduce LWD pressure and assist in restricting fungicides use, which has adverse risks to the environment and human health. Since different maize cultivars may react differently to such agrotechnical practices, each cultivar should be tested separately to determine its specific LWD protection suite. Field studies and long-term studies are needed to fully understand the potential of this approach.” (Lines 640-658)